# Drivers’ Decelerating Behaviors in Expressway Accident Segments under Different Speed Limit Schemes

**DOI:** 10.3390/ijerph19031590

**Published:** 2022-01-30

**Authors:** Wenhui Zhang, Jing Yi, Ge Zhou, Tuo Liu

**Affiliations:** School of Traffic and Transportation, Northeast Forestry University, Harbin 150040, China; zhangwenhui@nefu.edu.cn (W.Z.); lt375136225@nefu.edu.cn (T.L.)

**Keywords:** traffic accident segments, speed limit schemes, decelerating behavior, driving simulator

## Abstract

Traffic accidents occurring on expressways tend to give rise to traffic bottlenecks. To ensure the vehicles safely and smoothly pass through the accident segments, speed limits are generally taken to regulate the vehicles’ movements. This study aims to explore the decelerating behaviors of drivers under different speed limit schemes. We designed traffic accident scenarios under four speed limit schemes using the driving simulator. A total of 60 subjects drove the simulator passing the accident segments according to their habits. The vehicles’ kinematic data and the subjects’ operating data were recorded. To further analyze the drivers’ decelerating behaviors in different speed limit scenarios, driving experience was also taken into account. The results show that the speed limit schemes have significant effects on drivers’ decelerating behaviors. The more speed limit signs there are, the smoother the decelerating process will be. Driving experience significantly affects some of the decelerating parameters, including the location of deceleration starting point, average deceleration, and locations of decelerating to the initial and final speed limits. These results provide a theoretical basis for traffic safety and driving behavior management.

## 1. Introduction

Traffic safety has always been a hot issue in the field of transportation engineering. Approximately 200,000 traffic accidents occurred in China in recent years [1]. More than half of traffic accidents occur on expressways. Major accidents and extraordinarily serious accidents account for a relatively high proportion. After traffic accidents occur, the accident scenes tend to occupy part or all of the lanes for a long time. Traffic capacity, speed and safety on accident sections will, hence, suddenly decrease. Once the passing driver does not notice the traffic accident information, it is easy to cause secondary accidents, more serious casualties and property losses. According to relevant statistics, the number of secondary accidents on expressways accounts for 20% of all expressway traffic accidents. The fatalities from secondary accidents account for 18% of all expressway traffic fatalities [2].

In addition, the number of larger accidents caused by secondary accidents is as high as 60–80% and the number of casualties caused by secondary accidents is as high as 30–60% [3]. Compared to the traffic accidents without secondary collisions, the severity of traffic accidents with secondary collisions is 6.98 times larger [4].

Many studies have shown that excessive speed is one of the major factors to cause traffic accidents. A speed limit is a widely adopted measurement to improve traffic safety [5]. Speed limit signs showing the maximum or minimum speed value guide drivers to adjust vehicles’ speed. Especially under special climatic conditions or after traffic incidents’ occurrences, signs should be set at appropriate position in expressway. Too high or too low a speed limit value causes vehicles’ movements to not match the road or climatic conditions. The improper locations of signs also guide drivers’ misoperation.

There are many important factors that contribute to traffic accidents, including weather [6,7,8,9], road conditions [10], type of vehicle and its safety systems [11,12] and driving behavior [13], etc. According to previous research, about 84.84% of traffic accidents were related to driving behavior [14]. Deeply analyzing drivers’ operations helps us clearly understand and figure out drivers’ responses. Driving experience tend to be another important independent variable in the field of traffic safety [15]. Experienced drivers and novice drivers perform differently when facing traffic risks. Deceleration is the principal response to avoid crashes on accident segments, for most drivers [16]. However, different speed limit schemes for the same traffic accident segment may cause drivers to operate differently.

Most previous studies have focused on the comparison of speed limit values in expressway construction zones. The effect of speed limit on lane-changing behaviors on ideal lanes has also been studied. Less research has been conducted on the design of speed limit schemes and the analysis of decelerating behavior in expressway accident segments. In this study, a driving simulator was used to design accident scenarios with different numbers and layouts of speed limit signs. Then, the parameters of the drivers’ decelerating behaviors were analyzed. The changes in decelerating behaviors under different speed limit schemes and driving experience were obtained.

The main benefits of this study are that the designed speed limit schemes provide guidance on the setting of speed limit signs in expressway accident segments, which can effectively reduce the occurrence of secondary collisions. In addition, the decelerating parameters obtained in the present study can actively instruct the drivers to cope with the accidents and they can decelerate reasonably. Then, they can pass through the accident segments smoothly. This is of great significance to ensure traffic safety of expressway accident segments and reduce accident rates.

## 2. Literature Review

### 2.1. Safety Management in Expressway Accident Segments

Safety management measures for expressway accident segments include the use of changeable message signs or variable message signs (CMS or VMS), variable speed limit (VSL) and connected vehicle (CV) information interaction, which are adopted to reduce the risk of the secondary accidents. For example, Kopitch et al. [17] verified the effectiveness of 11 CMSs that provided information about accidents, work zone, congestion, speed limit and real time traffic to reduce the risk of accident segments. The effectiveness of the CMS increased between 2 mph and 11.15 mph and decreased between 11.15 mph and 22.3 mph. Lee et al. [18] observed subjects’ driving behaviors on the expressway with VSLS and VMS using a driving simulator. The warning messages displayed by the VMS and the VSL helped to reduce speed variation and traffic congestion. The risk of expressway crashes was decreased and the efficiency was increased. Gaweesh et al. [19] used a driving simulator to design a CV scenario and a non-CV scenario in bad weather. The vehicle kinematic data obtained from the experiments were analyzed. The CV improved the driving behaviors by decreasing the speed and speed variation and mitigated the risk of the secondary accidents for heavy trucks in expressway accident segments. Mishra et al. [20] developed an intelligent traffic fatality prevention system based on secure cloud (SeC) and the internet of things (IoT).

Some studies have also predicted that secondary accidents radically mitigate the occurrence of vehicle collisions in expressway accident segments. Sarker et al. [21] analyzed the accident data of Shelby County. A generalized ordered response probit (GORP) model was developed to predict the frequency of secondary crashes (SC) in expressway accident segments. The main influencing factors of the occurrence of SC were obtained. Xu et al. [22] analyzed the accident data from the 1-880 expressway, US and developed a model based on the logit model to predict the risk of secondary collisions. The impact variables of secondary crashes, including traffic volume, average speed, acceleration, etc., were obtained. The results showed that the developed model could effectively predict the risk of secondary crashes. The accuracy of this model had been improved compared to the previous prediction models.

### 2.2. Speed Limits in Expressway Accident Segments

Accident scenes usually form traffic bottlenecks on expressways, which can easily lead to secondary accidents. Speeding has become a major cause of traffic accidents [23]. To reduce the risk of secondary accidents, speed limits are generally adopted. Most studies have used variable speed limit (VSL) theory to explore the ways to reduce the risk of accident segments. For example, Li et al. [24] modified the vehicle tracking model to simulate the passage of vehicles passing the accident scenes under VSL control in bad weather. Results were obtained that VSL was effective in decreasing the risk of secondary collisions in all types of weather. Peng et al. [25] designed a combined controller using the VSL and lane change guidance (LCG) theories to avoid secondary collisions on accident segments.

Traffic bottlenecks formed in expressway accident segments can also lead to rear end collisions, and the VSL control can also be used to decrease the risk of rear end collisions. For example, Chen et al. [26] proposed a VSL scheme to improve the capacity of expressway bottlenecks caused by accident segments or work zones and ensured that drivers passed through the bottlenecks at smooth speeds. Wu et al. [27] developed a VSL control algorithm considering the visibility in foggy conditions. A feedback framework was developed to combine CV and VSL control. The results found that VSL played an important role in reducing the risk of rear end collisions. Xu et al. [28] proposed a procedure to help determine where VSL signs were set to lessen the risk of collisions when bottlenecks occurred in expressway accident segments.

Some studies have also been conducted to obtain calculated speed limit values for different accident scenes by analyzing the driver’s ability to deal with information. In turn, speed limit schemes are proposed and evaluated by microscopic traffic simulation. However, this method has not been studied deeply and comprehensively enough in relation to accident segments. It is mostly used for work zones, construction zones and reconstruction and extension zones. For example, Cheng et al. [29] proposed the methods of setting speed limit signs in the absence of traffic flow upstream of the work zones. When the design speed was 120 km/h, the speed limit values were set at 20 km/h and 10 km/h during daytime and nighttime, respectively, in the decelerating process. Finley et al. [30] proposed a bilevel programming model, combined with cases to obtain the number, location and values of VSL signs. A simulation was used to analyze the empirical characteristics of traffic accidents in expressway work zones. The setting of the VSL signs significantly reduced the traffic accidents in expressway work zones.

### 2.3. Drivers’ Decelerating Behaviors on Expressways

In order to lessen the risk of collisions of vehicles and traffic congestion, drivers often decelerate to avoid the risk. Some scholars used driving simulators to study the decelerating behaviors of drivers. For example, Domenichini et al. [31] assessed drivers’ decelerating behavior on the expressways with eight configurations, including speed limits and VSL, using a driving simulator, and collected data on drivers’ decelerating process. The results indicated that drivers generally drove at speeds above the speed limits, and they would decelerate only in bypasses. Decelerating behavior decreased the risk of collisions. Varotto et al. [32] analyzed the effects of VSL and automatic incident detection (AID) systems on drivers’ decelerating behaviors when approaching traffic congestion. When VSL and AID were present, the driver’s maximum deceleration was lowest when approaching congestion. VSL and AID could reduce traffic congestion and accident rates. Lyu et al. [33] analyzed the effects of driving experience, gender and occupation on the parameters of decelerating behaviors in deceleration lanes, such as speed, acceleration and the location of lane. Lyu et al. [34] analyzed driving differences in the deceleration lane between 46 subjects according to gender and driving experience.

The existing studies about decelerating behaviors on expressways have focused on the deceleration lanes of expressways to analyze the factors influencing decelerating behaviors and parameters. Although some studies have taken driving experience as an influencing factor, it was not common to take both driving experience and speed limits as independent variables to analyze decelerating behaviors. Speed limit management on expressways generally used variable speed limit signs; the design and analysis of speed limit schemes were not sufficiently in depth. Speed and acceleration were the main decelerating parameters analyzed in the existing studies, while trajectory parameters, such as the location of the deceleration starting point, have not been fully and thoroughly investigated.

## 3. Methodology

### 3.1. Apparatus

A 6-degree-of-freedom (6-DOF) driving simulator was used to conduct the experiments, as shown in Figure 1a. The driving simulator is mainly composed of a hardware system and software system. The hardware system includes a display screen, driving seat and sound system. The display screen is made up of 3 LCD screens, which provides a 135° field of view for drivers. The driving seat simulates a real seat with an adjustable and movable back. The angle data of the steering wheel can be read to achieve driving steering in the simulated scenarios. The driving simulator also sends signals to the force-feedback steering wheel through real time road conditions and speed to control the force feedback of the steering wheel and present the real force-feedback operation. The sound system is used as a 3D sound simulation system to generate the sound of all vehicles during driving, which consists of 1 subwoofer and 5 small speakers. The software system, including a bus framework, data processing system and simulation system, has powerful functions of simulation control and scene editing. The software interface for the run time scenario of the driving simulator is shown in Figure 1b.

The parameters related to the driver’s behavior (steering wheel corner, pedal opening, pedal force, time to pedal, etc.), the parameters related to vehicles (real time speed, distance, location, acceleration, etc.) and the parameters related to the road (radius of curve, slope, width) were recorded using the driving simulator. The driving data were automatically exported in the form of a spreadsheet of Microsoft Excel (Microsoft, Redmond, Washington, DC, USA, 2019). The driving simulator had a data collection frequency of 100 Hz; one set of data was generated every 0.01 s in the exported table.

### 3.2. Scenario

The scenarios were constructed referring to Sections K192+000 to K202+500 of Jianfu Expressway, China. This section was a two way dual carriageway with a total length of 10.5 km and a width of 3.75 m each lane. Taking a major traffic accident case as a prototype, a closed accident segment of 200 m in length was set up in the left hand lane.

Scenario with no speed limit sign (Scenario 1) and scenarios with one speed limit sign (Scenario 2), two speed limit signs (Scenario 3) and three speed limit signs (Scenario 4) were separately set up in front of the accident segment. According to the standard specification, the driving speed should be indicated on the expressway and the maximum speed should not be less than 60 km/h [35]. In general, 120 km/h is used to calculate the driving speed on expressways, and 100 km/h or 80 km/h is usually used to calculate the driving speed in specific conditions [36]. Therefore, 60 km/h was chosen as the minimum speed limit value in this study. It was more appropriate to choose 80 km/h as the maximum speed limit value, considering our accident segment. The average value of the maximum and minimum speeds was taken as the median speed. Therefore, 60 km/h, 70 km/h and 80 km/h had been determined as the three speed limit values.

These four scenarios were set up on the same section and every scenario was separated by 2000 m, as shown in Figure 2a–d. Ancillary facilities, such as guardrails, trees, light poles and so on, were set up on the roadside. The warning signs were set up 700 m in front of the accident areas, and every speed limit sign was separated by 200 m.

### 3.3. Subjects

The basic factors that should be considered when recruiting subjects include the overall number, age, driving experience and so on. To make the subjects have certain degree of representativeness, various factors must be considered comprehensively when recruiting. The subjects’ mental state and the presence or absence of physical diseases should also be confirmed. A total of 60 volunteers were recruited to participate in the experiments through online recruitment. The subjects were 30 experienced drivers ranging from 29 to 47 years old and 30 novice drivers ranging from 23 to 35 years old. Every experienced driver was identified as having held a driving license for more than 5 years with an average annual mileage of at least 8000 km. Novice drivers referred to those with less than 3 years of driving experience. All subjects had normal or corrected vision.

The required sample size is calculated based on the expected variance, target confidence level and margin of error. The equation for the sample size based on the power analysis is as follows:(1)$$n=\frac{{\left(z_{\alpha /2}+z_{\beta }\right)}^{2}\sigma^{2}}{\varepsilon^{2}}$$
where *n* is the sample size, *z* is the statistical value of the standard normal distribution. *α* and 1-*β*, respectively, represent the confidence level and the power of a test, *α* usually takes a value of 0.05 or 0.1 and *β* usually takes a value of 0.1 or 0.2. Furthermore, *σ* denotes the overall standard deviation of the normal distribution. When *σ* is unknown, it can be replaced by the sample standard deviation of *s*. Then, the corresponding test for the normal distribution is transformed into a test for the t distribution. When the sample size is large enough, the t distribution is approximately equal to the normal distribution. The above equation still holds. Furthermore, *ε* is the difference between the mean value of the response indicator and the mean value of the reference indicator, and the value of *ε* is often difficult to estimate. From the equation for the effect size, as shown in Equation (2), we can obtain the Equation (3) [37].
(2)δ=|ε|/σ,
(3)ε=±δσ

Equation (3) is substituted into Equation (1) to yield Equation (4).
(4)$$n=\frac{{\left(z_{\alpha /2}+z_{\beta }\right)}^{2}}{\delta^{2}}$$
where *δ* is the tolerable error. The interval for empirical values of *δ* is 0.25≤δ≤0.5 [38].

The variables of the above equations were taken as appropriate values, thus, the number of the required subjects were calculated. The experimental data of 56 subjects were, ultimately, valid. The basic details of the subjects are shown in Table 1.

### 3.4. Experimental Procedure

The test drives were conducted after programming scenarios using the driving simulator, aiming to verify the rationality of the scenarios. Then, the subjects were recruited for the experiments. The follow up specific experimental process is as follows:The subjects were asked to fill out a questionnaire about age, driver’s license validity and driving mileage after arriving at the driving simulation laboratory;A member of the research team introduced the driving simulator and the driving tasks to be completed by the subjects;The researchers guided the subjects in performing exercises before the experiments and ensured that they were familiar with the simulated environment and vehicle control. The experiments had to be stopped if the subjects were found to feel unwell during the experiments;The subjects were asked to formally carry out the driving tasks. The data were stored and named after the experiments.

The flow of the driving simulation experiment is shown in Figure 3.

### 3.5. Decelerating Parameters

The decelerating parameters were automatically collected by the driving simulator and later exported in the form of an Excel spreadsheet. To study the difference in the drivers’ decelerating behaviors on the accident segments under different speed limit schemes, the variable parameters related to the study needed to be filtered out from the exported data. The final filtered decelerating parameters are shown in Table 2.

To better explain the decelerating process and describe decelerating parameters, a driver’s driving process in Scenario 3 was taken as an example to plot the speed and acceleration curves during the decelerating process, as shown in Figure 4. Speed at the deceleration starting point (*_vDS_*), location of the deceleration starting point (*l_D_*_1_), location of decelerating to the initial speed limit (*l_D_*_2_), location of decelerating to the final speed limit (*l_D_*_3_), average deceleration (${\bar{a}}_{1}$) under different speed limit scenarios (*L*) and driving experience (*E*) were analyzed in this study. The speed limit scenarios included Scenario 1 with no speed limit sign (*L*_1_), Scenario 2 with 1 speed limit sign (*L*_2_), Scenario 3 with 2 speed limit signs (*L*_3_) and Scenario 4 with 3 speed limit signs (*L*_4_). Driving experience included novice drivers (*N*_1_) and experienced drivers (*E*_1_).

The 5 variables during the decelerating process were interpreted as follows:

Speed at the deceleration starting point (*v_DS_*) (km/h): the speed corresponding to the deceleration starting point;Location of the deceleration starting point (*l_D_*_1_) (m): The difference between the location coordinate of the first speed limit sign and the location coordinate corresponding to the deceleration starting point. For example, −400 minus the abscissa corresponding to point P1, which is generally a positive value;Location of decelerating to the initial speed limit (*l_D_*_2_) (m): the difference between the location coordinate of the first speed limit sign and the location coordinate of decelerating to the initial speed limit;Location of decelerating to the final speed limit (*l_D_*_3_) (m): The difference between the location coordinate of the last speed limit sign and the location coordinate of decelerating to the final speed limit. For example, −200 minus the abscissa corresponding to point P3;Average deceleration (${\bar{a}}_{1}$) (m/s^2^): The average deceleration from the deceleration starting point to the final speed limit. For example, the average value of deceleration during the process from point P1 to point P3.

## 4. Results

### 4.1. Speed Distribution of the Upstream and Downstream of the Accident Segments

The trends in the overall speed of drivers who are 1000 m in front of the starting point of the accident segment for four accident scenarios are shown in Figure 5. The abscissas represent the location coordinates of the expressway segments on which the vehicles are driving. The abscissas of point 0 correspond to the location of the starting point of the accident segment. The ordinates represent the real time speed of the drivers on the expressway segments. To observe the trends and patterns in drivers’ speed in different scenarios, the locations of the speed limit signs were marked with vertical lines of different color and the starting points of the accident segments were marked with red dashed lines.

The drivers’ speeds remained stable before seeing the speed limit signs. Then, they began to decelerate after seeing the speed limit signs. By comparing the changes in drivers’ speed in different scenarios, the characteristics were initially obtained as follows:Figure 5a showed that the drivers remained at a basically stable speed in the range of 80 km/h to 120 km/h at 500 m away from the accident segment when there was no speed limit sign. When driving to the location in the range of 200 m to 500 m away from the accident segment, they had varying degrees of decelerating behaviors;From Figure 5b–d, the drivers’ speeds remained stable before seeing the speed limit signs. A total of 90% of drivers maintained speeds in the range of 90 km/h to 120 km/h. There were certain differences in drivers’ speed from the time they saw the speed limit signs to the time they started to decelerate;The settings of the speed limit sign in different scenarios had significant impacts on drivers’ speed control. From Scenario 2 to Scenario 4, the speed profile of the drivers gradually flattened out. It could be seen that, as the speed limit signs increased, the drivers would decelerate gently in advance;From Figure 5b–d, the drivers’ decelerating behaviors were earlier when there were more speed limit signs.

### 4.2. Speed at the Deceleration Starting Point

Statistical methods were used to prove that different speed limit signs have impacts on the drivers’ speed in different locations. Previous studies have shown that driving experience has an impact on driving behavior. Therefore, analysis of variance (ANOVA) is used to analyze the effects of speed limit scenarios and driving experience on speed, acceleration and location.

The box plot of the speed at the deceleration starting point considering different speed limit scenarios and driving experience is shown in Figure 6. It shows certain differences between the trends in the speed at the deceleration starting points for different speed limit scenarios.

As shown in Table 3, the analysis of variance showed that speed at the deceleration starting point was not influenced by driving experience (*p* > 0.05) but was significantly influenced by speed limit scenarios (*F* (2,165) = 0.874, *p* < 0.001). In conjunction with Figure 6, the speed at the deceleration starting point gradually decreased as the number of the speed limit signs increased. In addition, the interaction between driving experience and speed limit scenarios had no significant effect on the speed at the deceleration starting point (*p* > 0.05).

Table 4 listed the multiple test results of the speed at the deceleration starting point in different scenarios. The difference in the speed at the deceleration starting point was not significant between Scenario 3 and Scenario 4 (*p* > 0.05), but it was significant between Scenario 2 and Scenario 3 as well as between Scenario 2 and Scenario 4. In addition, the difference in the speed at the deceleration starting point between Scenario 2 and Scenario 4 was the largest (6 km/h).

### 4.3. Location of the Deceleration Starting Point

Figure 7 showed that the location of the deceleration starting point of experienced drivers were greater than that of novice drivers. Novice and experienced drivers had the same trend in the location of the deceleration starting point in all three scenarios. This result indicated that the experienced drivers would generally decelerate earlier than the novice drivers when seeing the speed limit signs.

As shown in Table 5, the analysis of variance revealed the significant impacts of driving experience (*F* (1,166) = 35.737, *p* < 0.001) and speed limit scenarios (*F* (2,165) = 6.960, *p* = 0.001) on the location of the deceleration starting point. Furthermore, the interaction between driving experience and speed limit scenarios had no statistically significant effect on the location of the deceleration starting point (*p* > 0.05).

As shown in Table 6, the difference in the location of the deceleration starting point between Scenario 2 and Scenario 4 was not significant (*p* > 0.05). The significance of the differences in the location of the deceleration starting point were all larger between Scenario 2 and Scenario 3 as well as between Scenario 2 and Scenario 4. The difference in the location of the deceleration starting point between Scenario 2 and Scenario 3 was the largest, reaching 128 km.

### 4.4. Deceleration

Excessively small deceleration may prevent the drivers from decreasing their speeds to the specified values in time. This situation causes the drivers to break into the accident segment. Therefore, it is necessary to study the differences in the deceleration in different scenarios. Taking the driving process of a subject as an example, the acceleration curves of the three scenarios are plotted in Figure 8. The variation in acceleration in Scenario 2 (*L*_2_) was the largest, and the second largest variation in acceleration appeared in Scenario 3 (*L*_3_). The acceleration changed most gently in Scenario 4 (*L*_4_). This showed that the drivers would decelerate more gently when more speed limit signs were in place. The drivers started to decelerate at least 500 m in front of the speed limit sign in Scenario 2. However, they began to decelerate within 200 m of the speed limit signs in Scenario 3 and Scenario 4. This was consistent with the analysis results of the location of the deceleration starting point.

Figure 9 indicated that, as the number of speed limit signs increased, the absolute values of the average deceleration gradually decreased. The absolute values of the average deceleration for experienced drivers were smaller than that for novice drivers. Thus, the distributions of the average deceleration for novice drivers were more discrete than that for experienced drivers. In addition, the average deceleration for Scenario 2 had a more discrete distribution than that for Scenarios 3 and Scenario 4. This result showed that the more speed limit signs there were, the smoother the drivers’ decelerating process would be. In addition, the experienced drivers would decelerate more gently than the novice drivers.

As shown in Table 7, the analysis of variance revealed the significant impacts of driving experience (*F* (1,166) = 4.786, *p* = 0.03) and speed limit scenarios (*F* (2,165) = 21.136, *p* < 0.001) on the average deceleration. Furthermore, the effect of the interaction between driving experience and speed limit scenarios on the average deceleration was significant (*F* (2,165) = 4.419, *p* = 0.014).

Table 8 showed that the significance of the differences in the average deceleration between each of the three scenarios was relatively larger. The difference in the average deceleration between Scenario 2 and Scenario 4 was the largest, reaching −1.234 m/s^2^.

### 4.5. Location of Decelerating to the Initial Speed Limit

#### 4.5.1. Drivers’ Compliance with the Speed Limits

The drivers’ ability to decrease the speed to the specified value in a timely and smooth manner with the guidance of the speed limit signs not only reflect the drivers’ operating level but also whether the speed limit signs are set up reasonably. By comparing the drivers’ ability to cross the location of the speed limit signs when decelerating to the initial speed limit, the drivers’ compliance with the speed limit signs in different scenarios is obtained, as shown in Table 9. The number of speed limit signs is varied in different scenarios. In this study, the location of decelerating to the initial speed limit refers to the distance between the location of reaching to the initial speed limit and the location of the speed limit signs. Therefore, this makes the behaviors of drivers comparable in all scenarios. There are five sets of noncompliant data in the three scenarios. The number of times that the drivers decreased the speed to the specified value after driving past the speed limit signs in Scenario 3 (*L*_3_) was the most, reaching three times. In Scenario 4 (*L*_4_), the number of times that the speed limit signs were not obeyed was the fewest.

The reason why the drivers did not obey the speed limit signs may be the relatively fewer number of traffic signs set up in Scenario 2 (*L*_2_) and Scenario 3 (*L*_3_), and the speed was relatively faster. Therefore, the drivers did not have sufficient buffer space and speed during the decelerating process. Ultimately, the decelerating process was not completed. It could be seen that the experienced drivers who did not obey the speed limit signs were relatively more. This phenomenon showed that experienced drivers did not urgently change the deceleration to reach the specified speed limit value, and they were more likely to decelerate more gently. Actually, when the drivers are driving at fast speeds, their decelerations should not be too large even if the target value for decelerating is relatively lower. Excessive deceleration not only causes loss of control of the vehicles, but also rear end accidents and other traffic accidents.

#### 4.5.2. Drivers’ Location when Decelerating to the Initial Speed Limit

Figure 10 showed the hotspot distribution map of the location of decelerating to the initial speed limit in three scenarios. When there was only one speed limit sign, the distribution color of the location of decelerating to the initial speed limit was mostly at depths of 128–198 m. When there were two speed limit signs, the distribution color of the location of decelerating to the initial speed limit was mostly at depths of 128–296 m. When there were three speed limit signs, the distribution color of the location of decelerating to the initial speed limit was mostly at depths of 198–230 m.

In Figure 11, the location of decelerating to the initial speed limit of novice drivers was greater than that of experienced drivers. As the number of speed limit signs increased, the location of decelerating to the initial speed limit of novice and experienced drivers in the three scenarios showed a downward trend.

As shown in Table 10, the analysis of variance revealed the significant impacts of driving experience (*F* (1,160) = 5.127, *p* = 0.025) and speed limit scenarios (*F* (2,159) = 12.579, *p* < 0.001) on the location of decelerating to the initial speed limit. However, the effect of the interaction between driving experience and speed limit scenarios on the location of decelerating to the initial speed limit was not significant (F (2,165) = 0.172, *p* = 0.842).

Table 11 indicated that the differences in the location of decelerating to the initial speed limit had larger significances between Scenario 4 and Scenario 2 as well as between Scenario 4 and Scenario 3 (*p* = 0.011, *p* = 0.001), but the difference in the location of decelerating to the initial speed limit between Scenario 2 and Scenario 3 was not significant (*p* > 0.05). The difference in the location of decelerating to the initial speed limit between Scenario 2 and Scenario 4 was the largest, reaching 52 m.

### 4.6. Location of Decelerating to the Final Speed Limit

Since there is only one speed limit sign in Scenario 2 (*L*_2_), it makes sense to analyze the location of decelerating to the initial speed limit as well as the location of decelerating to the final speed limit. The location of decelerating to the initial speed limit for Scenario 2 has been analyzed in this study. Therefore, the location of decelerating to the final speed limit for Scenario 3 and Scenario 4 will be analyzed.

In Figure 12, when there were two speed limit signs, the color distribution of the location of decelerating to the final speed limit was wide. This also meant that the distribution of the location of decelerating to the final speed limit was more discrete, and it was mostly at depths of 159–310 m. When there were three speed limit signs, the color distribution of the location of decelerating to the final speed limit was relatively smaller, which was mostly at depths of 8–159 m. This was because the drivers would maintain lower speeds when there were more speed limit signs. Thus, the driving environment was relatively safer at that time. The drivers were relaxed when driving at low speeds and they would decelerate to the prescribed speed limit later when seeing the final speed limit sign.

Figure 13 showed that the location of decelerating to the final speed limit of novice drivers was greater and more discrete than that of experienced drivers. The location of decelerating to the final speed limit of Scenario 3 (*L*_3_) was greater than that of Scenario 4 (*L*_4_).

As shown in Table 12, the analysis of variance revealed the significant impacts of driving experience (*F* (1,110) = 25.508, *p* < 0.001) and speed limit scenarios (*F* (2,109) = 6.138, *p* = 0.015) on the location of decelerating to the final speed limit. However, the interaction between driving experience and speed limit scenarios had no significant impact on the location of decelerating to the final speed limit (*F* (2,109) = 0.443, *p* = 0.507).

## 5. Discussion

### 5.1. Analysis of Speed Limit Schemes

The speed limit schemes proposed in this study include a scenario with no speed limit sign and scenarios with a single speed limit sign and multiple speed limit signs. According to the Chinese standard specifications, the speed limit values were selected as 60 km/h, 70 km/h and 80 km/h. The design of the speed limit schemes was also related to the scenario selection of the driving simulator used in this study.

Following the study conducted by Filtness et al. [39], a driving simulator was also used to design the speed limit scenarios. The speed selected for the speed limit schemes was 100 km/h. Moreover, Filtness et al. [39] emphasized using a driving simulator to study the impact of the location of the speed limit signs on driving safety, which was slightly different from the content of this study.

However, the design of the speed limit schemes and recruitment of subjects in this study are the same as the study conducted by Yang et al. [40]. Therefore, the driving simulating experiments and the speed limit schemes are reasonable.

In real life applications, when an accident occurs on an expressway, the speed limit schemes proposed in the present study can be referred to temporarily set up the speed limit signs in front of the accident segment. Alternatively, the locations and values of the speed limit signs can also be temporarily changed on the basis of the original speed limit signs of the expressway. This application can effectively reduce the occurrence of secondary accidents.

### 5.2. Analysis of Decelerating Behavior

Driving experience had no significant effect on the speed at deceleration starting point. However, the speed limit scheme was an important influencing factor. The interaction between driving experience and speed limit schemes was irrelevant to the speed at deceleration starting point. The results were the same as the conclusions obtained by Yang et al. [40], but they believed that the speed at deceleration starting point was susceptible to the distance between the construction areas. As the conditions set by this study are different from theirs, the conclusions are somewhat different.

For the location of deceleration starting point, both driving experience and speed limit schemes were important influencing factors. However, their interaction had little influence. The location of the deceleration starting point of experienced drivers was greater than that of novice drivers, which was similar to the results received by Yang et al. [40].

Deceleration was also affected by driving experience and speed limit schemes. As the number of speed limit signs continued to increase, the deceleration continuously decreased. The deceleration of experienced drivers was lower than that of novice drivers, which was the same as the conclusions obtained by Yang et al. [40].

The deceleration, location of decelerating to the initial speed limit and location of decelerating to the final speed limit of novice drivers were all higher than that of experienced drivers. This result was similar to Yang et al. [40], but they focused on the reaction time, and the steering and braking parameters of novice and experienced drivers when avoiding a collision. The setting of speed limit schemes in the driving simulating experiment conducted by Yang et al. [40] is different from ours, so the results are also slightly different.

Experienced drivers mostly disobey the speed limit signs in expressway accident segments, but most novice drivers basically obey the speed limit signs. This is similar to the conclusions obtained by Zovak et al. [5]. They believed that drivers especially obey the speed limits in tunnels, and only a certain number of drivers obey the speed limits on expressways.

We conducted driving simulating experiments and designed the simulating scenarios considering the accident intensity, driving experience and traffic flow. The methodology has reference value and the observed data is beneficial for researchers all over the world.

In fact, in addition to driving experience, weather, road conditions, gender, type of vehicles, time of accident, location of accident, width of accident scene, etc., are all main factors influencing driving behavior. It may be useful to assess decelerating behaviors by considering these factors. Therefore, these factors will be added to the future studies.

Furthermore, the results related to the decelerating behaviors obtained in this paper can be practically applied to the safety trainings of drivers. For novice drivers, the appropriate location, time, speed, etc., for decelerating obtained in the present study can be used as a reference to instruct them to pass the expressway accident segment and they can also be guided on how to judge and obey the speed limit signs. This training should be given to novice drivers when they obtain driving licenses. For experienced drivers, they should be urged to obey the speed limit signs with reasonable incentives and penalties. Although they have some experience in responding to expressway accidents, they should also be advised on the appropriate location, time, speed, etc., for decelerating.

## 6. Conclusions

Driving simulating experiments were conducted to design the accident scenarios under different speed limit schemes to analyze drivers’ decelerating behaviors. The following conclusions were drawn based on the analysis of the relationship between various parameters of the decelerating process and driving experience and different speed limit schemes, which were positively meaningful for speed limit management and guiding vehicles to pass through the accident segment.

Speed limit scheme had a significant impact on all parameters of the decelerating process in the expressway accident segment. The more speed limit signs there were, the lower the speed at the deceleration starting point and average deceleration for drivers would be. Then, the decelerating process became smoother. Therefore, multiple speed limit signs (i.e., a step by step speed limit) could be used to reduce the average deceleration on accident segments. Thus, drivers could decelerate in advance. Since drivers’ speeds on expressways were generally between 90 km/h and 120 km/h [35], the absolute values of drivers’ average deceleration should not be higher than 1 m/s^2^, in order to avoid accidents. The absolute values of the average deceleration for Scenario 2 and Scenario 3 were greater than that for Scenario 4. By analyzing the locations of decelerating to the initial and final speed limits, it could be seen that the drivers would generally obey the speed limits in Scenario 4 (with three speed limit signs). The distributions of the locations of decelerating to the initial and final speed limits were separately at depths of 30–198 m and 8–159 m. Additionally, in this scenario, they could drive through the accident segment more gently and calmly. This shows that the speed limit effect for Scenario 4 is the best. Therefore, traffic management signs can be set up with reference to the drivers’ decelerating indicators of Scenario 4.

Driving experience had a significant impact on the location of deceleration starting point, average deceleration, and the locations of decelerating to the initial and final speed limits. The location of deceleration starting point for experienced drivers was 100–200 m ahead of that for novice drivers. The absolute values of the average deceleration for experienced drivers were approximately 0.5 m/s^2^ smaller than that for novice drivers. The location of decelerating to the initial and final speed limits for novice drivers were larger than that for experienced drivers. The difference in the location of decelerating to the initial speed limit was within 100 m but in the location of decelerating to the final speed limit was within 200 m. Thus, it could be concluded that experienced drivers usually decelerate earlier than novice drivers when seeing the speed limit signs. The decelerating process of experienced drivers was smoother than that of novice drivers. Although many experienced drivers did not obey the speed limit signs, they would decelerate more gently rather than urgently decelerate to increase their safety. They tended to obey the signs with the increasing of the speed limit signs. In contrast, novice drivers were more likely to actively obey the speed limit signs due to their lack of the driving experience, and they are slightly impatient in decelerating.

In summary, these results provided quantitative support for the settings of the speed limit signs, and the parameters of drivers’ decelerating behaviors obtained in this study could guide both experienced and novice drivers in passing through the accident segments. Therefore, a number of countermeasures can be proposed to improve level of traffic safety based on above results, which are as follows:Installation of step by step speed limit signs in expressway accident segments. Speed limit signs of 60 km/h, 70 km/h and 80 km/h can be, respectively, placed 200 m, 400 m and 600 m in front of the accident segments to ensure that drivers can decelerate in time according to the speed limit values before approaching the accident segments, which also plays a warning role. The probability of secondary accidents can be decreased and drivers’ safety can be improved;Management of driving behaviors in expressway accident segments. When there is no speed limit sign, drivers are advised to decelerate 200–500 m in front of the accident segments. When there are speed limit signs, drivers are advised to decelerate when seeing the signs. However, experienced drivers should be advised to decelerate 100–200 m earlier than novice drivers. To ensure a smooth decelerating process, drivers’ average deceleration should not be higher than 1 m/s^2^ and drivers’ speed should preferably be kept at 90 km/h before decelerating;Training of the driver’s operation. Drivers can be recruited to carry out simulated operation using a driving simulator. It is possible to check if the parameters of the driving behavior are within safety limits and the operation is standardized. If so, the drivers’ operation is proven to be safe, otherwise the drivers’ operation needs to be corrected and trained until it is up to the safety standard.

However, this study has certain limitations. The accident scenarios are designed to be excessively simple. When the simulated accident scenarios are designed, the accident segments on other types of lanes are not considered. The factors influencing driving behaviors, such as bad weather, road conditions and gender, are not added into the simulated scenarios. In a future study, a two way six lane carriageway or two-way eight-lane carriageway can be considered when designing accident scenarios. Scenarios under rainy, snowy and foggy weather can also be simulated. In addition, gender should be considered when recruiting the subjects. Different proportions of men and women can be recruited to participate in experiments to analyze the differences in driving behaviors between men and women.

## Figures and Tables

**Figure 1 ijerph-19-01590-f001:**
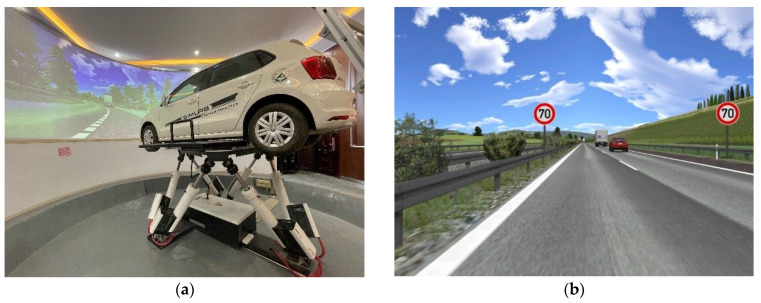
Driving simulator: (**a**) 6-DOF driving simulator; (**b**) the scenario of driving simulating software.

**Figure 2 ijerph-19-01590-f002:**
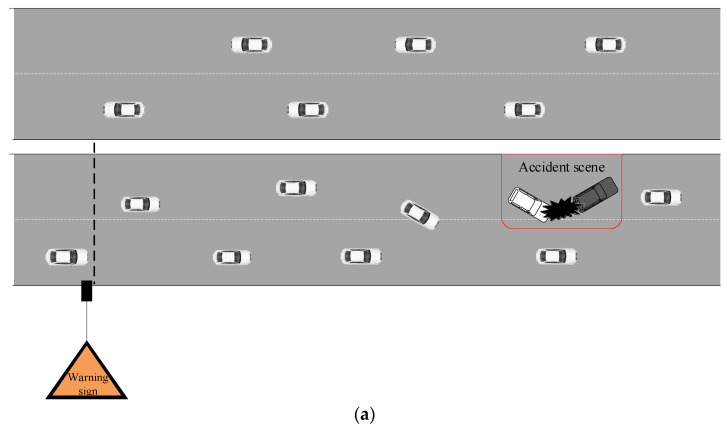
Four experimental scenarios: (**a**) Scenario 1; (**b**) Scenario 2; (**c**) Scenario 3; (**d**) Scenario 4.

**Figure 3 ijerph-19-01590-f003:**
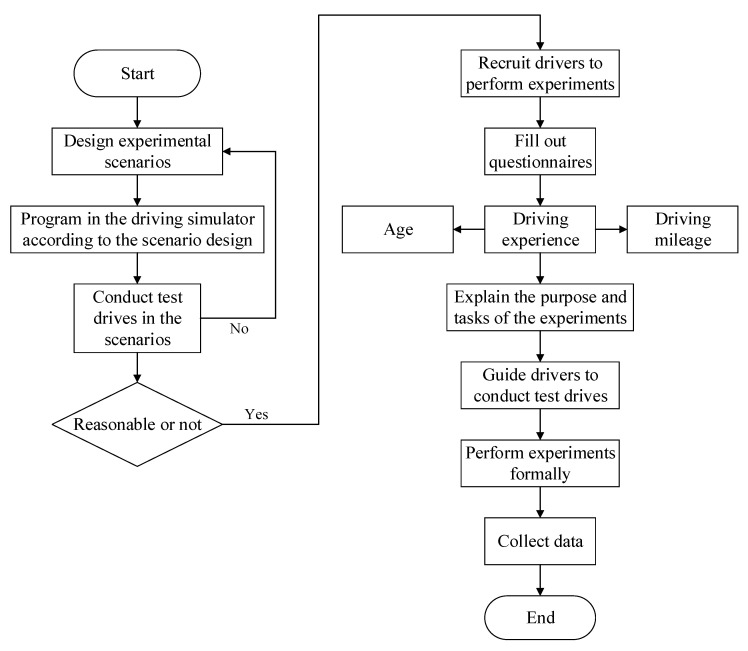
Experimental flowchart.

**Figure 4 ijerph-19-01590-f004:**
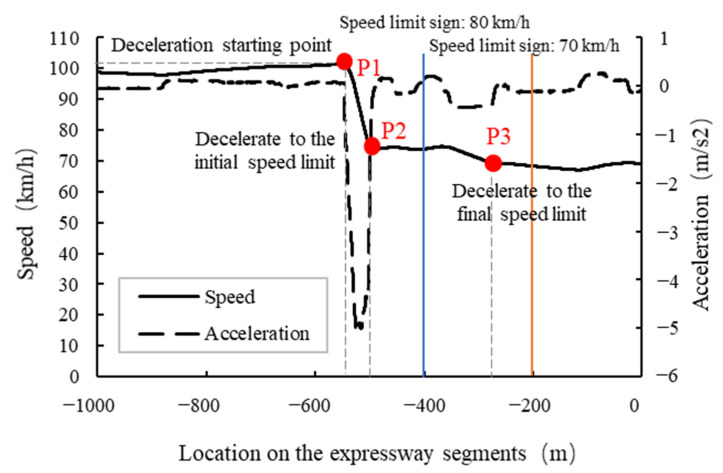
Driver’s decelerating process (Scenario 3).

**Figure 5 ijerph-19-01590-f005:**
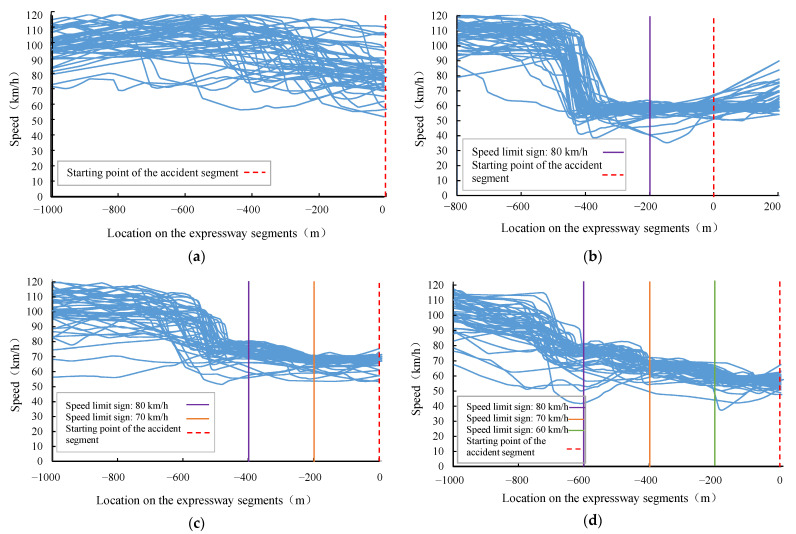
Changes in speed of vehicles passing through the accident segment: (**a**) Scenario 1; (**b**) Scenario 2; (**c**) Scenario 3; (**d**) Scenario 4.

**Figure 6 ijerph-19-01590-f006:**
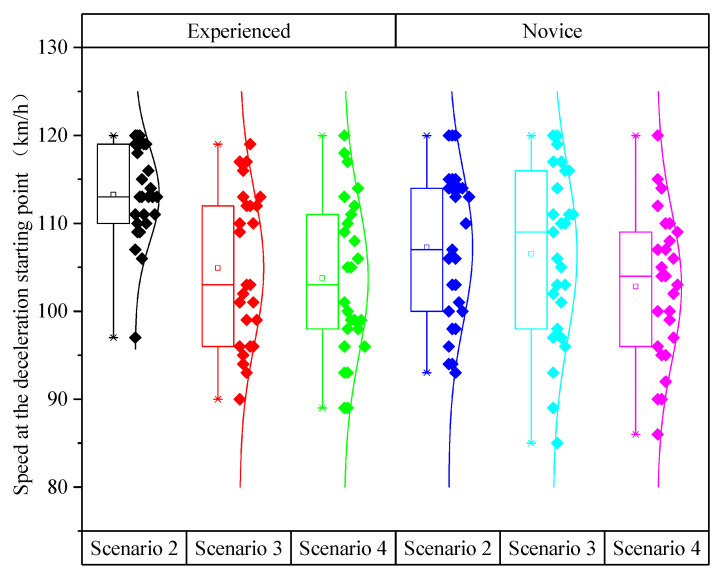
Speed at the deceleration starting points under different scenarios and driving experience.

**Figure 7 ijerph-19-01590-f007:**
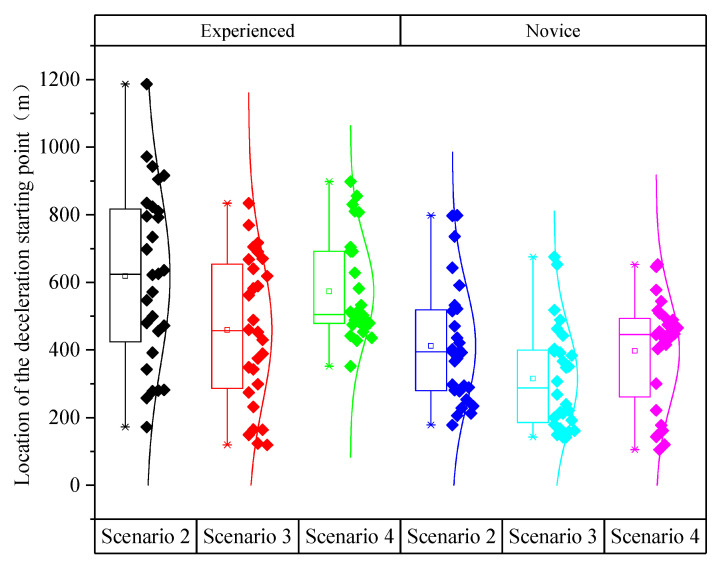
Location of the deceleration starting point under different scenarios and driving experience.

**Figure 8 ijerph-19-01590-f008:**
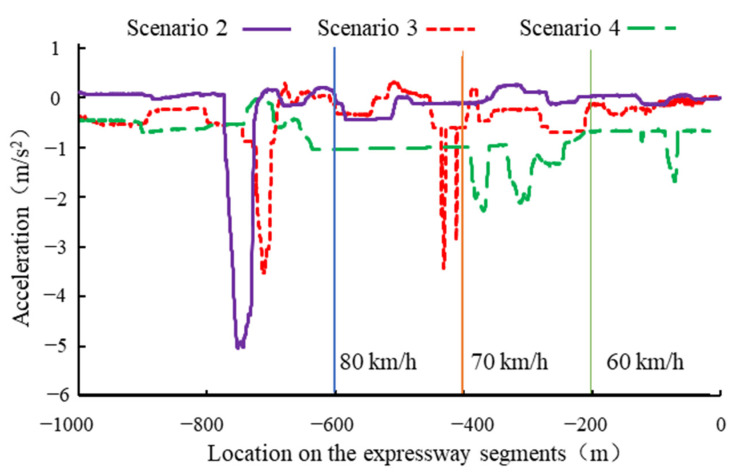
Acceleration curves during decelerating process.

**Figure 9 ijerph-19-01590-f009:**
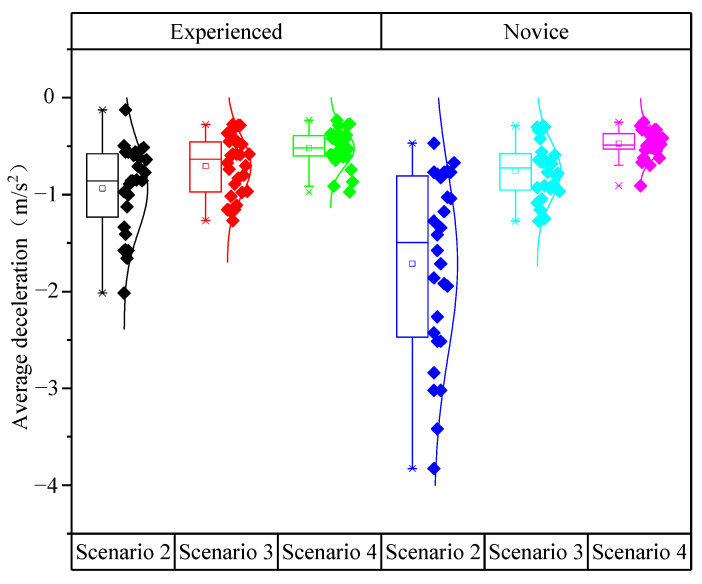
Distribution of the average deceleration for experienced and novice drivers.

**Figure 10 ijerph-19-01590-f010:**
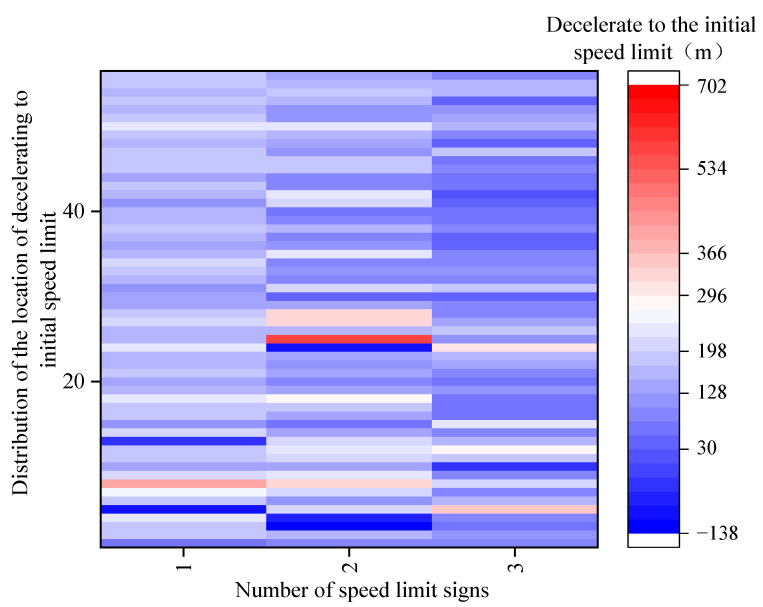
Distribution of the location of decelerating to the initial speed limit in different scenarios.

**Figure 11 ijerph-19-01590-f011:**
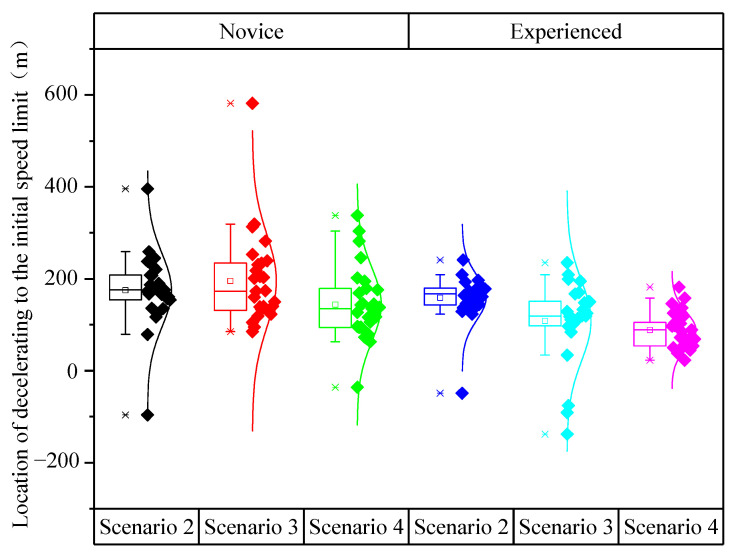
Location of decelerating to the initial speed limit under different scenarios and driving experience.

**Figure 12 ijerph-19-01590-f012:**
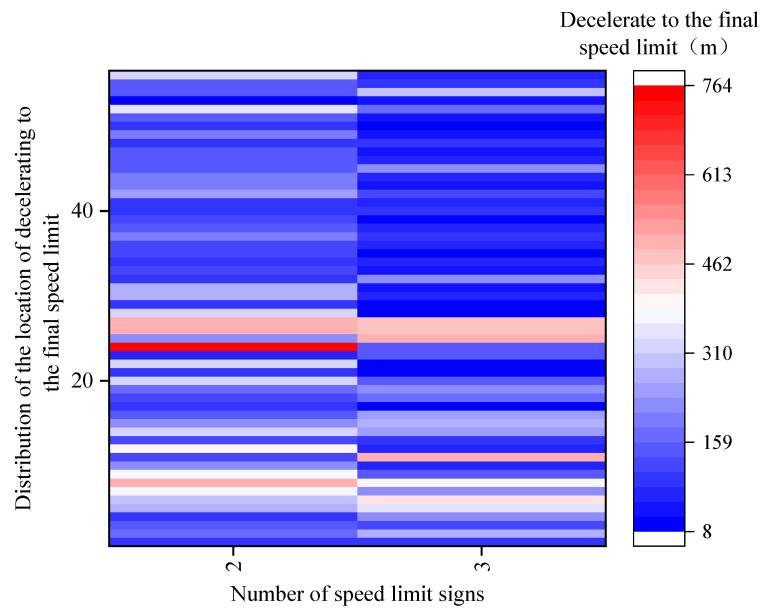
Distribution of the location of decelerating to the final speed limit in different scenarios.

**Figure 13 ijerph-19-01590-f013:**
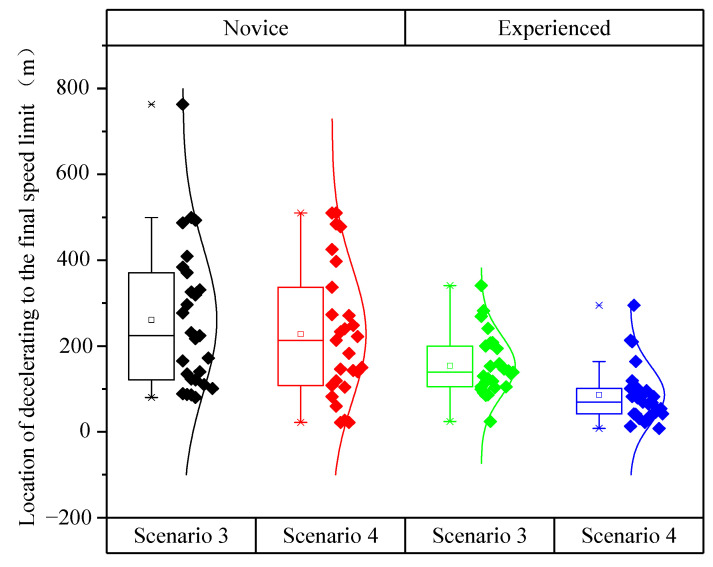
Location of decelerating to the final speed limit under different scenarios and driving experience.

**Table 1 ijerph-19-01590-t001:** Basic information about the subjects.

Driver Type	Novice	Experienced	Total
Male	21	24	45
Female	7	4	11
Total	28	28	56

**Table 2 ijerph-19-01590-t002:** Filtered decelerating parameters.

Data Classification	Decelerating Parameters
Speed (km/h)	Driving speed before accident area; speed at the deceleration starting point;
Acceleration (m/s^2^)	Acceleration in the decelerating process
Vehicle trajectory (m)	Location of the deceleration starting point; location of decelerating to the initial speed limit; location of decelerating to the final speed limit

Note: Speed at the deceleration starting point and location of the deceleration starting point mean that the most recent speed begins to decrease continuously before the drivers approach the first speed limit sign. At this time, the corresponding location and speed are the location of the deceleration starting point and speed at the deceleration starting point. The location of decelerating to the initial speed limit refers to the location where the most recent speed decrease to the initial speed limit when approaching the first speed limit sign. The same is true for location of decelerating to the final speed limit.

**Table 3 ijerph-19-01590-t003:** Results of the analysis of variance for the speed at the deceleration starting point.

Variables	Class III Sum of Squares	df	Square of the Mean	F	Significance
*E*	87.414	1	87.414	1.200	0.275
*L*	1271.753	2	635.877	8.731	0.000
*E*L*	323.605	2	161.802	2.222	0.112

**Table 4 ijerph-19-01590-t004:** Multiple test results of the speed at the deceleration starting point.

Scenarios	Scenarios	Average Difference	Standard Error	Significance	95% Confidence Interval of Difference	
					Lower Limit	Upper Limit
*L* _3_	*L* _4_	2.704	1.642	0.102	−0.540	5.948
	*L* _2_	−4.111 *	1.642	0.013	−7.355	−0.867
*L* _4_	*L* _3_	−2.704	1.642	0.102	−5.948	0.540
	*L* _2_	−6.815 *	1.642	0.000	−10.059	−3.571
*L* _2_	*L* _3_	4.111 *	1.642	0.013	0.867	7.355
	*L* _4_	6.815 *	1.642	0.000	3.571	10.059

* Significance level of 0.01.

**Table 5 ijerph-19-01590-t005:** Results of the analysis of variance for the location of the deceleration starting point.

Variables	Class III Sum of Squares	df	Square of the Mean	F	Significance
*E*	1,278,736.006	1	1,278,736.006	35.737	0.000
*L*	498,118.583	2	249,059.292	6.960	0.001
*E*L*	27,793.655	2	13,896.827	0.388	0.679

**Table 6 ijerph-19-01590-t006:** Multiple test results of the location of the deceleration starting point.

Scenarios	Scenarios	Average Difference	Standard Error	Significance	95% Confidence Interval of Difference	
					Lower Limit	Upper Limit
*L* _3_	*L* _4_	−96.429 *	35.748	0.008	−167.021	−25.836
	*L* _2_	−128.018 *	35.748	0.000	−198.610	−57.426
*L* _4_	*L* _3_	96.429 *	35.748	0.008	25.836	167.021
	*L* _2_	−31.589	35.748	0.378	−102.182	39.003
*L* _2_	*L* _3_	128.018 *	35.748	0.000	57.426	198.610
	*L* _4_	31.589	35.748	0.378	−39.003	102.182

* Significance level of 0.01.

**Table 7 ijerph-19-01590-t007:** Results of the analysis of variance for the average deceleration.

Variables	Class III Sum of Squares	df	Square of the Mean	F	Significance
*E*	4.901	1	4.901	4.786	0.030
*L*	43.289	2	21.644	21.136	0.000
*E*L*	9.050	2	4.525	4.419	0.014

**Table 8 ijerph-19-01590-t008:** Multiple test results of the average deceleration.

Scenarios	Scenarios	Average Difference	Standard Error	Significance	95% Confidence Interval of Difference	
					Lower Limit	Upper Limit
*L* _3_	*L* _4_	−0.486 *	0.191	0.012	−0.864	−0.108
	*L* _2_	0.748 *	0.191	0.000	0.370	1.126
*L* _4_	*L* _3_	0.486 *	0.191	0.012	0.108	0.864
	*L* _2_	1.234 *	0.191	0.000	0.857	1.612
*L* _2_	*L* _3_	−0.748 *	0.191	0.000	−1.126	−0.370
	*L* _4_	−1.234 *	0.191	0.000	−1.612	−0.857

* Significance level of 0.01.

**Table 9 ijerph-19-01590-t009:** Drivers’ compliance with the speed limits in different speed limit scenarios.

Driver Type	*L* _2_		*L* _3_		*L* _4_	
	Compliance	Noncompliance	Compliance	Noncompliance	Compliance	Noncompliance
*N* _1_	54	1	53	3	55	1
*E* _1_	56	1	56	0	56	0
total	110	2	109	3	111	1

**Table 10 ijerph-19-01590-t010:** Results of the analysis of variance for the location of decelerating to the initial speed limit.

Variables	Class III Sum of Squares	df	Square of the Mean	F	Significance
*E*	26,373.389	1	26,373.389	5.127	0.025
*L*	129,409.864	2	64,704.932	12.579	0.000
*E*L*	1765.444	2	882.722	0.172	0.842

**Table 11 ijerph-19-01590-t011:** Multiple test results of the location of decelerating to the initial speed limit.

Scenarios	Scenarios	Average Difference	Standard Error	Significance	95% Confidence Interval of Difference	
					Lower Limit	Upper Limit
*L* _3_	*L* _4_	39.018 *	15.144	0.011	9.113	68.922
	*L* _2_	−13.018	15.144	0.391	−42.922	16.887
*L* _4_	*L* _3_	−39.018 *	15.144	0.011	−68.922	−9.113
	*L* _2_	−52.036 *	15.144	0.001	−81.940	−22.131
*L* _2_	*L* _3_	13.018	15.144	0.391	−16.887	42.922
	*L* _4_	52.036 *	15.144	0.001	22.131	81.940

* Significance level of 0.01.

**Table 12 ijerph-19-01590-t012:** Results of the analysis of variance for the location of decelerating to the final speed limit.

Variables	Class III Sum of Squares	df	Square of the Mean	F	Significance
*E*	394,368.893	1	394,368.893	25.508	0.000
*L*	94,889.286	1	94,889.286	6.138	0.015
*E*L*	6851.571	1	6851.571	0.443	0.507

## Data Availability

The data used to support the findings of this study are available from the corresponding author upon request.

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
