# Peer review of "Drivers’ Decelerating Behaviors in Expressway Accident Segments under Different Speed Limit Schemes"

_ijerph, 2022, doi:10.3390/ijerph19031590_

Round 1

Reviewer 1 Report

The paper submitted is interesting and it may be helpful for relevant audience. In the paper, expressway accident segments are considered and some speed limits are applied. The aim of the study is to explore the decelerating behaviors of drivers under different speed limit schemes and traffic accident scenarios. The driving simulator is used for this purpose. 

Although the article is generally well written ; however, there are some issues and statements in the paper that need revisions. My concerns and suggestions are as follows:

1. The main benefits of this study should be explained. 

2. The outcomes of the study should be discussed in details regarding the real life applications.

3. According to the results of this study what kind of countermeasures can be taken into consideration to improve level of traffic safety. This issue should be regarded.

Reviewer 2 Report

There are many abbreviations used in the text and figure captions, which makes it difficult to read the article (ANOVA LD1 VDs etc.).

Reviewer 3 Report

Document with comments from the reviewer is attached

Round 2

Reviewer 3 Report

After examining the changes applied, as well as the feedback from the authors, we have verified that the authors have carefully considered the suggestions made by us.
We hope that the changes made to the article have been an improvement for it and for its readers.